# Epicatechin Inhibited Lipid Oxidation and Protein Lipoxidation in a Fish Oil-Fortified Dairy Mimicking System

**DOI:** 10.3390/foods12071559

**Published:** 2023-04-06

**Authors:** Zhenghao Lian, Jiahui Han, Yue Cao, Wenhua Yao, Xiaoying Niu, Mingfeng Xu, Jun Xu, Qin Zhu

**Affiliations:** 1Key Laboratory for Quality and Safety of Agricultural Products of Hangzhou City, College of Life and Environmental Sciences, Hangzhou Normal University, Hangzhou 311121, China; lianzhenghao@stu.hznu.edu.cn (Z.L.); 2021111010051@stu.hznu.edu.cn (J.H.); tumiantuuuuu@gmail.com (Y.C.); yaowenhua1@stu.hznu.edu.cn (W.Y.); niuxiaoying9946@gmail.com (X.N.); zjxmf@163.com (M.X.); 2Jiaxing Key Laboratory of Preparation and Application of Advanced Materials for Energy Conservation and Emission Reduction, School of Advanced Materials & Engineering, Jiaxing Nanhu University, 572 South Yuexiu Road, Jiaxing 314001, China; xujun@jxnhu.edu.cn

**Keywords:** lipid oxidation, protein lipoxidation, epicatechin, whey protein isolate

## Abstract

In this study, a typical tea polyphenol epicatechin (EC) was investigated for its impact on the oxidative stability of whey protein isolate (WPI) in a fish oil-fortified emulsion. The oil-in-water emulsion system consisted of fish oil (1%, *w*/*w*), WPI (6 mg/mL), and EC (0.1, 1, and 2 mM), and the oxidation reaction was catalyzed by Fenton’s reagent at 25 °C for 24 h. The results showed EC exhibited a dose-dependent activity in the reduction of lipid oxidation (TBARS) and protein carbonylation. A Western blot analysis demonstrated that protein lipoxidation was inhibited by EC via interrupting the covalent binding of lipid secondary oxidation products, MDA, onto proteins. In addition, protein lipoxidation induced a loss of tryptophan fluorescence, and protein hydrolysis was partially recovered by EC. The findings of this study provide an in-depth understanding of the performance of phenolic antioxidants in relieving lipid oxidation and subsequent protein lipoxidation in oil-containing dairy products.

## 1. Introduction

The nutritional benefits of higher amounts of polyunsaturated fatty acids (PUFAs), mainly ω-3 PUFAs, promote the development of PUFAs-fortified food products. Fish oil, microalgae oil, or flaxseed oil are currently the main source of ω-3 PUFAs utilized in dairy-based foods [1]. However, the incorporation of PUFAs into food matrices is hindered by its oxidation sensitivity [2]. The double bonds in unsaturated fatty acid chains are highly susceptible to oxidative attack. Oxidation reactions produce primary peroxides (i.e., lipid hydroperoxides), and these lipid peroxides further decompose into a broad range of reactive carbonyl species (RCS) as secondary oxidation products, such as malondialdehyde (MDA), 4-hydroxynonenal (4-HNE), acrolein (ACR), and glyoxal (GO), etc. [3]. In the food system, the amino acids in proteins are targets of these RCS. The reactions leading to the covalent attachment of RCS to proteins were named “protein lipoxidation” with the formation of advanced lipid oxidation end products (ALEs). Generally, there are two common reactions between RCS and proteins: (i) a reaction with protein amino groups in a hydrophobic manner to form Schiff bases and (ii) a reaction with nucleophiles in the form of a Michael addition to form carbon–carbon double bonds [4].

RCS-induced protein lipoxidation leads to structural modifications imposed on amino acids and changes in protein physicochemical and functional properties, formation of toxic compounds, and a possible transfer of oxidative damage from food proteins to physiological proteins or the human microbiota. With the dietary intake, RCS and ALEs will be absorbed through the gastrointestinal tract into lymph or blood stream. Their ingestion and accumulation are risk factors contributing to the development of atherosclerosis and several types of cancers, especially the colon cancer [3]. Concerning the adverse effects of lipid oxidation and protein lipoxidation reactions in food quality and human health, effective antioxidant measures are applied to prevent and control the progress of oxidation. The application of natural polyphenols as potential antioxidants has been widely accepted in the food industry [5]. Polyphenols are extensively present in the nature as secondary metabolites of plants, mostly found in fruits and vegetables [6]. Among them, epicatechin (EC), which is found in green tea, cocoa, berries, grape seeds, and apples, is considered one of the most efficient flavonols in inhibiting oxidation reactions in oil-in-water emulsions [7]. EC has the typical C6-C3-C6 skeleton, and the B ring structure contains *ortho*-dihydroxyl groups, which are considered as outstanding antioxidants by scavenging free radicals or chelating metal ions. Meanwhile, a resorcinol structure of the A ring has high nucleophilic centers for trapping of lipid oxidation-derived RCS [8]. However, the occurrence of hydroxyl groups in phenolic antioxidants is often accompanied by multiple chemical reactions with other food components. A mix of both covalent (formation of phenol–protein compounds) and non-covalent (hydrophobic interactions, hydrogen binding, and ionic binding) interactions between polyphenols and proteins was found [8,9]. These interactions might exert influence on protein’s molecular configuration, depending on types and dosage of polyphenols employed [9,10]. In a previous study, tea polyphenols were demonstrated to inhibit both lipid and protein lipoxidation at a lower concentration (100 mg/L) while the promotion of protein carbonylation sulfhydryl loss was found at higher concentration (400 mg/L) of tea polyphenols [10]. Therefore, the level of polyphenolic antioxidants should be optimized to avoid the introduction of unwanted quality changes in foods.

The aim of this work is to elucidate the inhibitory effects of EC on lipid oxidation and protein lipoxidation in a fish oil-fortified dairy mimicking system containing WPI (Whey protein isolate). WPI contains mainly β-lactoglobulin, α-lactalbumin, bovine serum albumin, immunoglobulin, lactoferrin, and the consumption of WPI has been shown to enhance protein synthesis for elder persons, effectively improving muscle performance and preventing muscle atrophy [11]. Most of the WPI products in the market are more or less fortified with other nutritional supplements, such as fish oil, to achieve a balanced nutritional profile. Therefore, reinforcing the oxidative stability by adding antioxidants is of great importance to these products. In this dairy mimicking system, lipid oxidation was monitored by thiobarbituric acid (TBA) assay while the influence of oxidation on protein physicochemical properties was revealed by the changes in protein carbonylation, sulfhydryl content, intrinsic tryptophan, and surface hydrophobicity. Direct evidence of protein lipoxidation was obtained with Western blot analysis on the identification of MDA-bound proteins in WPI. In addition, changes in protein surface microstructure and protein hydrolysis were evaluated. The results of this study may contribute to a better understanding of oxidation-induced damage to dairy proteins in emulsion-based foods enriched with PUFAs and facilitate the potential application of natural polyphenols in controlling or preventing oxidation in emulsion systems.

## 2. Materials and Methods

### 2.1. Materials

Pure fish oil was obtained from Zhejiang Shenzhou Marine Bioengineering Co., Ltd. (Zhoushan, Zhejiang, China). The composition of fatty acids (*w*/*w*) was determined by GC-MS and the total PUFAs content was 31.16%. WPI (protein content 92.9%) was commercial product of Hilmar Ingredients (Hilmar, CA, USA). Epicatechin (EC), 2,4-dinitrophenylhydrazine (DNPH), trichloroacetic acid (TCA), thiobarbituric acid (TBA), phosphate-buffered saline (PBS), phosphate-buffered saline (PBS), brilliant blue R250, 1,1,3,3-tetramethoxypropane, guanidine hydrochloride, and 5,5′-dithiobis (2-nitrobenzoic acid) were the products of Tokyo Chemical Industry Co., Ltd. (Tokyo, Japan) or Sinopharm Chemical Reagent Co., Ltd. (Shanghai, China). 1,1,3,3-tetramethoxypropane, porcine pepsin (250 units/mg solid), and trypsin (8 USP size) were the products of Sigma-Aldrich Co. LLC (St. Louis, MO, USA). The rabbit polyclonal antibody to malondialdehyde-modified proteins (ab27642) and its corresponding secondary antibody was provided by Abcam (Cambridge, UK).

### 2.2. Emulsion Preparation

Oil-in-water emulsion containing fish oil and whey proteins was prepared according to the method of Obando et al. [12]. First, fish oil and WPI were dissolved in phosphate buffer saline (20 mM, pH 7.4) with the addition of EC, and the mixtures were homogenized with a high-speed blender (XFK FSH-2B, Changzhou, Chinese) at 10,000 rpm for 2 min. Final concentrations of the components in the emulsion were 1% (*w*/*v*) of fish oil, 6 mg/mL of WPI and 0, 0.1, 1, 2 mM of EC. Fenton oxidation reaction mixture consisted of FeCl_3_ (10 μM), L-ascorbic acid (100 μM), and H_2_O_2_ (1 mM). Oxidation reaction was carried out in the dark for 24 h at 25 °C with constant shaking in a thermostatic shaker (TS-100B, Shanghai TianCheng Experimential instrument Manufacturing Co., Ltd. Shanghai, China). An amount of 0.05% sodium azide was utilized to suppress the growth of microbes. Trolox was added immediately to terminate the reaction after 24 h incubation. WPI control group is the emulsion containing only WPI; WPI fish oil group is the emulsion containing WPI co-oxidized with fish oil; and WPI fish oil EC groups are emulsions containing WPI co-oxidized with fish oil in the presence of EC (0.1, 1, 2 mM).

### 2.3. Analyses of Lipid Oxidation

The degree of lipid oxidation was determined by TBARS (Thiobarbituric acid reactive substances) assay [13]. Briefly, the sample (0.3 mL) was mixed with an equal amount of TBA reagent (dissolving 15% TCA and 0.375% TBA in 2 M HCl). The mixture was then heated in a water bath at 95 °C for 30 min. After cooling to room temperature (25 °C) and centrifugation (6000× *g* for 5 min), the absorbance of pinkish supernatant was monitored at 532 nm by a Varioskan™ microplate reader (Thermo Scientific, Waltham, MA, USA). MDA was obtained by the hydrolysis of TMP (1,1,3,3-tetramethoxypropane) and TBARS value (μM) was calculated by the construction of a TBA-MDA standard curve.

### 2.4. Analyses of Protein Carbonylation

Protein carbonyl content was determined according to the procedure of Levine et al. [14]. An amount of 0.2 mL DNPH derivatization reagent (0.1% *w*/*v*, in 2 M HCl) was mixed with 0.1 mL of each sample. The derivatization was carried out for 1 h in the dark at room temperature. An amount of 0.2 mL of 20% TCA solution (*w*/*v*) was then added to precipitate the protein, and the mixture was centrifuged at 6000× *g* for 5 min. The precipitate was washed three times with 0.2 mL of ethanol/ethyl acetate (1:1, *v*/*v*) solution. The resultant precipitate was finally dissolved in 0.6 mL of 8 M guanidine hydrochloride. Protein carbonyl content (nmol carbonyl per mg protein) was measured spectrophotometrically at the UV absorbance of 370 nm for protein hydrazones and calculated by the absorption coefficient of 22,000 mol^−1^cm^−1^.

### 2.5. Total Sulfhydryl Groups

The content of protein-bound thiols in different treatments were evaluated using Ellman’s method [15]. Briefly, 0.2 mL urea-SDS solution (8.0 M urea and 30 mg/mL SDS in 0.1 M phosphate, pH 7.4) was added to 0.1 mL of each sample. The mixture reacted with 10 mM 5,5′-dithiobis (2-nitrobenzoic acid) at room temperature for 30 min. After reaction, the mixture was centrifuged (6000× *g* for 5 min), and the supernatant was measured colorimetrically at 412 nm. A molar extinction coefficient of 13,600 mol^−1^cm^−1^ was adopted for the calculation of total sulfhydryl content, and the results are expressed as mmol/g protein.

### 2.6. Intrinsic Tryptophan Fluorescence

The impact of EC on intrinsic tryptophan fluorescence in WPI under oxidative stress was measured and analyzed according to the method of Wu et al. [16]. Briefly, each emulsion sample (3 mL) was mixed with an equal volume of 20% TCA solution (*w*/*v*) to precipitate the protein, and the mixture was centrifuged at 5000× *g* for 5 min. The precipitated proteins were re-dissolved in 3 mL PBS before the intrinsic fluorescence intensities were recorded by a Hitachi-F4600 model fluorescence spectrometer at an excitation wavelength of 285 nm, an emission wavelength of 300~400 nm, and a slit width of 5 nm.

### 2.7. Surface Hydrophobicity

Sodium 8-aniline-1-naphthalenesulfonate (ANS) was used as a fluorescent probe for protein surface hydrophobicity according to the procedure of Li et al. [17]. Briefly, each sample was diluted to a range of concentrations from 0.1 to 0.5 mg/mL with PBS, and 3 μL of ANS reagent was added to each diluted sample (0.2 mL), and the fluorescence intensity were recorded at an excitation wavelength of 365 nm and an emission wavelength of 484 nm. The protein surface hydrophobicity index was expressed as the slope of linear regression of fluorescence intensity versus protein concentration (mg/mL).

### 2.8. Measurement of Protein Hydrolysis

The degree of hydrolysis (DH) of the proteins was measured by o-phthaldialdehyde (OPA) method according to the procedure of Church et al. [18]. OPA reagent was prepared by mixing of 0.1% SDS (*w*/*v*), 0.08% OPA (*w*/*v*), and 0.088% dithiothreitol (*w*/*v*) in 2% ethanol (*v*/*v*). The sample was first adjusted to pH 1.5 with HCl, and pepsin was added, and gastric phase digestion was carried out at 37 °C. After 1 h of digestion, pH was adjusted to 7 with NaOH, and pepsin was added to initiate intestinal digestion at 37 °C for 2 h. Aliquots 30 μL were taken and mixed with 600 μL OPA reagent at different time points (30, 60, 90, and 180 min) during digestion for the measurement of DH. Absorbance was measured at 340 nm immediately after a 37 °C water bath for 2 min. A standard curve was constructed with 0~10 mM serine, and free amino content was calculated from the standard curve for estimation of the degree of hydrolysis. A = 1 and β = 0.4 for WPI. The formula was as follows:$$\begin{array}{c} W_{serine-NH_{2}}=C_{serine-NH_{2}}\times \frac{V\cdot N}{X\cdot P\%}\\ h=\frac{W_{serine-NH_{2}}-\beta }{\alpha }\\ \mathrm{DH}=\frac{h}{h_{tot}} \end{array}$$
where $W_{serine-NH_{2}}$ is the amount of serine-NH_2_ per gram of protein; X (g) is the mass of the sample; P% is the mass fraction of protein in the sample; V (L) is the volume of the sample; N is the dilution factor of the sample; h (mmol/g) is the number of peptide bonds broken per gram of protein during digestion during the hydrolysis process; h_tot_ (mmol/g) is the total number of peptide bonds per gram of protein; the h_tot_ of WPI is 8.8; and α and β are represented by constants 1 and 0.4, respectively.

### 2.9. SDS-PAGE Analysis

SDS-PAGE was used to analyze the distribution of molecular weight of WPI proteins. In general, each protein sample (3 μL) was heated and processed by mixing with 3 μL loading buffer (4×) containing β-mercaptoethanol. The samples were denatured for 5 min at 95 °C before loading onto a 5% polyacrylamide stacking gel. Proteins were separated on a 15% polyacrylamide resolving gel under a voltage of 200 V. The gels after protein separation were stained with Coomassie Brilliant Blue R250.

### 2.10. Western Blot Analysis

The protein molecules were separated by gel electrophoresis and then transferred to a PVDF membrane for subsequent immunoblotting. Non-specific binding was blocked by 0.05 g/mL non-fat milk overnight at 4 °C. The membrane was washed 4 times with PBS-Tween (PBS buffer containing 0.05% Tween 20, pH 7.4) before incubation with a 1:5000 dilution (*v*/*v*) of primary anti-antibody (anti-MDA, ab27642, Abcam) for 3 h at 25 °C. The membrane was washed 3 times and incubated with a corresponding secondary HRP-conjugated antibody for 1 h. Chemilluminescence images of MDA-bound protein in WPI was finally visualized in a Kodak X-ray film using a Pierce visualizer spray & glow ECL Western Blot detection kit (Thermo Fisher, Waltham, MA, USA).

### 2.11. Scanning Electron Microscope (SEM) Observation

The surface microstructure of WPI was observed using a high-resolution S-4800 SEM (Hitachi Co., Tokyo, Japan). Specimens were fixed on conductive gel and coated with gold. The microstructure was observed and photographed at a magnification of 3000 times under an accelerating voltage of 3 kV.

### 2.12. Data Analyses

All experiments were run in triplicates, and data was reported as mean and standard deviation using SPSS software for data analysis. Duncan’s multiple extreme difference test was used for analysis of variance (ANOVA) with a significance threshold of 5%.

## 3. Results and Analysis

### 3.1. Lipid Oxidation

The oxidative stability of fish oil was assessed by formation of lipid oxidation secondary products using TBARS assay. As shown in Figure 1A, when compared to the control (0.03 μM), the oxidation products of fish oil developed rapidly as the TBARS reached a maximum value of 0.88 μM in oil-in-water emulsion. TBARS value declined dose-dependently to 0.40, 0.31, and 0.21 μM in the presence of EC (0.1, 1, and 2 mM), respectively. Similar to the present study, the prevention of lipid oxidation by phenolic compounds (black rice anthocyanins, rosemary extract, and green tea polyphenols) was observed in oil-containing emulsion systems [7,19,20]. The *ortho*-dihydroxyl groups in B ring of EC play a vital role in antioxidative activities via neutralizing free radicals via donating electron(s) or chelating transition metal irons [21]. The inhibition of the lipid oxidation by EC might contributed to the mitigation of protein lipoxidation initiated by lipid oxidation products.

### 3.2. Protein Carbonylation

Since carbonylated amino acids are not intrinsic constituents of native proteins, these highly reactive groups must be introduced under oxidative stress [22]. RCS were derived from lipid oxidation, and an increase in protein carbonyl content is also one of the most important signs of protein lipoxidation. Changes in protein carbonyls are depicted in Figure 1B; the carbonyl content of native WPI was 0.47 μmol/g, which was significantly lower than that of 1.58 μmol/g protein in the presence of oxidized fish oil (*p* < 0.05). The carbonyl content decreased to 1.27 μmol/g protein with the addition of EC (0.1 mM) and further reduced to 0.80 and 0.69 μmol/g protein when the concentration of EC was 1 and 2 mM, respectively. Protein carbonyls in dairy proteins were generally produced through several pathways (i.e., via direct carbonylation of amino acid side chains or indirect modification by lipid oxidation products or glycation/glycoxidation products) [23]. In our study, the existence of 1% fish oil led to lipid oxidation, and the development of TBARS and protein carbonyls was timely coupled. Hence, it is highly unlikely that the lipid oxidation and protein lipoxidation take place independently in the oil-in-water emulsions. Presumably, the inhibition of protein carbonylation by EC might be attributed to its interruption of lipid oxidation with a lowered generation of carbonyl compounds as a precursor of protein carbonylation. However, the dose-dependent antioxidant activities of tea polyphenols have been not observed in walnut oil-in-water emulsions. While a lower concentration (100 mg/L) of polyphenols was effective in inhibiting protein carbonylation, high concentration (400 mg/L) of polyphenols promoted the formation of protein-bound carbonyls [10]. The molecular mechanism of these controversial effects is still unclear. Other influencing factors include environmental pH and the presence of transition metal chelators. Therefore, the dosage of tea polyphenols must be optimized before the application in O/W emulsions according to their type, concentration, localization, and molecular environment [9].

### 3.3. Intrinsic Fluorescence Changes

The fluorophores in tryptophan, tyrosine, and phenylalanine residues are regarded as the main source of endogenous fluorescence in proteins, and the loss of tryptophan fluorescence is one of the most common markers of protein modification under oxidative stress [24]. The fluorescence profile of WPI with different treatments was shown in Figure 2, the unoxidized WPI control has a maximum fluorescence emission wavelength (λmax) at 330 nm when excited at the wavelength of 280 nm. Intrinsic fluorescence intensity sharply declined under oxidative stress with a red shift of the absorption peak. A similar phenomenon was also seen in oxidized β-lactoglobulin under the oxidative stress of H_2_O_2_ [25] and WPI treated by the lipid oxidation product MDA [26]. The λmax of tryptophan located inside the protein is roughly 330 nm while the λmax of tryptophan located on the surface of the protein corresponds to a maximum absorption wavelength of roughly 345 nm [27]. The oxidation of tryptophan in nucleophilic side chains and the interaction between tryptophan and other molecules changed the structure and location of tryptophan residues and contribute to the red shift of λmax.

The addition of EC led to a dose-dependent recovery of the loss in fluorescence intensity. A possible explanation is the alleviation of lipid oxidation and protein carbonylation by EC, with lesser structural modification on tryptophan. However, decrease in protein intrinsic fluorescence intensity and red shifted λmax were observed in O/W emulsions mixed with certain polyphenols due to covalent or non-covalent binding between polyphenolic compounds and proteins [28]. In this study, on the contrary, the recovery of intrinsic fluorescence was observed in the presence of EC, suggesting the protective effects of EC in lipoxidation-induced impairment on protein intrinsic fluorescence.

### 3.4. Protein Sulfhydryl Content

During protein lipoxidation, the carbonyl groups are not the only oxidized sites. Other important structures, such as Cys residues, are also sensitive to oxidation reaction. The content and distribution of sulfhydryl and disulfide bond groups were changeable according to the redox state of Cys and the equilibrium constant of the sulfhydryl-disulfide bond exchange reaction. As a result, the quantification of protein sulfhydryl groups is a valuable means to assess protein damage [29]. As expected, in our study, the sulfhydryl content in the non-oxidized WPI was 30.7 nmol/mg protein, and its level significantly decreased to 18.4 nmol/mg protein when WPI was oxidized together with fish oil (Figure 3A). The loss of protein sulfhydryl content can be attributed to the alteration of protein hooks, which resulted in the formation of disulfide bonds [30]. In the presence of EC at the dosage of 0.1 and 1 mM, the sulfhydryl increased to 23.8 and 26.9 nmol/mg protein, respectively. Higher doses of EC promoted the reduction of sulfhydryl content to 21.3 nmol/mg protein. A plausible explanation for this is the role of EC as a pro-oxidant at high doses, which might lead to the depletion of protein sulfhydryls. It is commonly accepted that in an oxidizing environment, phenolic compounds can be oxidized to electrophilic quinones, which further react with nucleophilic groups of proteins (C-N or C-S) via a Michael addition, and non-covalent binding is also found to be involved in the interaction between polyphenols (chlorogenic acid, ferulic acid, and epigallocatechin-3-gallate) and β-lactoglobulin [28,31].

### 3.5. Protein Surface Hydrophobicity

Surface hydrophobicity reflects the number of hydrophobic groups exposed on protein surfaces, and it considerably affects the stability of proteins in O/W emulsions. In this study, the surface hydrophobicity of the complexes was assessed using ANS as a fluorescent probe [32]. As shown in the Figure 3B, the surface hydrophobicity of oxidized WPI (124.9) was significantly reduced compared to unoxidized WPI (157.0). The hydrophobic groups originally exposed on the protein surface lost their ANS binding sites due to changes in the protein tertiary structure and oxidative damage [33]. The binding of lipid oxidation derived products (e.g., hydroxyl radicals, hydroperoxides, and carbonyl groups, etc.) to the non-polar regions of WPI might contribute to the drop in protein surface hydrophobicity. The effects of 0.1 mM and 1 mM EC on hydrophobicity were not significant compared to oxidized WPI, but a higher dose of EC (2 mM) triggered a distinct loss of surface hydrophobicity on WPI. Similar to the above results for protein sulfhydryl groups, the change in hydrophobicity was more likely the result of protein–quinone production. In conclusion, the conformation of protein could be affected by lipoxidation and the interaction between EC and WPI, leading to an increase in the disordered structure, thus causing some degree of molecular stretch and exposure of internal hydrophobic groups and conversion of hydrophilicity to hydrophobicity.

### 3.6. Protein Hydrolysis

Based on the reaction of OPA with primary amines, the OPA spectrophotometric assay is a rapid, convenient, and sensitive method for the measurement of proteolysis in dairy proteins. The degree of protein hydrolysis (DH) is expressed as the ratio of the number of hydrolyzed peptide bonds to the total number of peptide bonds, which represents protein hydrolysis. As shown in Table 1, at the end of digestion, a sharp decrease to 8.22% of DH was observed after total gastrointestinal digestion while the DH of unoxidized WPI was 13.92%, suggesting the oxidized WPI was less accessible for proteases. The impact of oxidation on protein hydrolysis has been investigated in several previous studies and protein modification, and cross-linking and aggregation are thought to be responsible for changed digestive behavior, especially in the presence of PUFAs, which are more prone to oxidation [12,34]. In this study, a gradual increase in DH of WPI was found in EC-incorporated emulsions. When the concentration of EC reached 2 mM, the DH at the end of digestion was 14.95%, which was not significantly different from unoxidized WPI. The recovery of DH might be attributed to a progressively reduced degree of lipid and protein lipoxidation in the presence of EC. The result of protein digestibility was in accordance with Western blot analysis of protein modification and SEM observation of protein microstructure, indicating the protective effects of EC on the protein lipoxidation.

### 3.7. SDS-PAGE Analysis

SDS-PAGE was used to examine the impacts of oxidation on the molecular weight distribution of WPI. Electrophoresis results showed that two major proteins, β-lactoglobulin (β-Lg, 18.4 KDa) and α-lactalbumin (α-La, 14.2 KDa), were detected in WPI (Figure 4). A comparison of the protein profiles in unoxidized WPI and oxidized WPI revealed that the impact of oxidative damage on WPI was mainly found in both β-Lg and α-La. The band of β-Lg in oxidized WPI slightly shifted upward, indicating an increase in molecular weight, presumably due to covalent binding of small molecule lipid oxidation products to β-Lg. The band of α-La significantly faded, suggesting α-La is more susceptible to modification, even led to its fragmentation or degradation. Similarly, sharply decreased band intensity in α-La in oxidatively damaged WPI in the presence of fish oil and walnut oil has been observed [20]. Oxidation-induced protein modification was also revealed by the accumulation of larger protein aggregates, which cannot migrate across the stacking gel. The oxidative damage of α-La was partially alleviated by EC (0.1, 1, and 2 mM) as the band intensity gradually recovered. The above studies and our results suggest that cross-linking, polymerization, and degradation of proteins are unavoidable during oxidative damage.

### 3.8. Western Blot Analysis

Lipid oxidation-derived RCS could exert oxidative modifications on dietary proteins in various food systems [22,35]. MDA is one of the major secondary oxidation products of ω-3/ω-6 fatty acid oxidation, comprising 70% of the total aldehydes [36]. Proteins are spontaneously attacked by highly reactive MDA at the N-termini of peptides and nucleophilic ε-amino groups on amino acid side-chains (MDA-modified amino acid residues, predominantly MDA-lysine adducts), with the generation of fluorescent dihydropyridine (DHP)-type products, Schiff base adducts, and cross-links in proteins [31]. In our study, a highly specific MDA antibody was employed to monitor the formation of MDA-induced protein modification in WPI (Figure 5). As expected, no MDA-bound protein was detected in a control group when WPI was incubated without fish oil. MDA-bound proteins were observed in WPI when co-oxidized with fish oil, and the most intensive immunoreactivity was noticed in the band of β-Lg, suggesting β-Lg was the preferential targets of MDA in WPI. The existence of MDA-modified WPI in this study also indicated that protein modification takes place by attachment of reactive carbonyls to the protein. In combination with the slightly upward shift of the bands of β-Lg and α-La as visualized in SDS-PAGE electropherogram, it is assumed that higher molecular-weight adducts and intermolecular cross-linking were generated from lipid oxidation-derived carbonyl compounds, such as MDA. The Western blot analysis also confirmed the potency of EC to reduce the formation of MDA-bound proteins, as the immunoreactivity of the bands was appreciably depressed in a dose-dependent manner. This result is in agreement with the results of TBARs and protein carbonylation tests, as well as reported in previous studies suggesting tea polyphenols as effective agents in maintaining protein oxidative stability in O/W emulsions [37,38]. The underlying mechanism might be attributed to the antioxidant properties of EC in reducing the production of MDA. An alternative explanation is the nucleophilic property of tea polyphenols in direct reaction with MDA as an electrophile. MDA was preferentially reacting with tea polyphenols with lowered level of precursor in the formation of MDA-bound proteins [32,37,39].

### 3.9. Morphological Examination

Attempts were made to visualize the changes on protein surface morphology by scanning electron microscopy. It can be seen in Figure 6 that the WPI particles in the control group were small and uniformly distributed, with smooth surfaces and clear boundaries. Under oxidative stress, the WPI particles underwent obvious aggregation, with the formation of large, irregular polymers. It is speculated that the attack of lipid oxidation products on protein side chains may have led to protein cross-linking and protein aggregation [40]. This is consistent with the appearance of larger protein particles under oxidative stress in this study. The incorporation of EC did not restore protein microstructure to its unoxidized state as protein aggregation still existed. Besides protein cross-linking and protein aggregation, the interactions between polyphenols and proteins also contributed to the disrupted protein surface morphology.

## 4. Conclusions

Owing to the possibilities for multiple oxidative reactions in oil-in-water system, a comprehensive evaluation on both lipid oxidation and protein lipoxidation with the addition of phenolic antioxidant was performed in this study. Our results showed that tea polyphenol EC significantly inhibited TBARS formation and protein carbonylation. A Western blot analysis indicted an interruption of the binding of MDA to a protein was responsible for the attenuated protein lipoxidation. Meanwhile, at high concentration of EC (up to 2 mM), the polyphenol–protein interactions could contribute to the physicochemical property changes in WPI (sulfhydryl groups, surface hydrophobicity, protein hydrolysis, and surface morphology). Taken together, the addition of polyphenolic antioxidants to dairy products for protein lipoxidation is theoretically feasible, and the effect of the dosage on the protein itself cannot be ignored.

## Figures and Tables

**Figure 1 foods-12-01559-f001:**
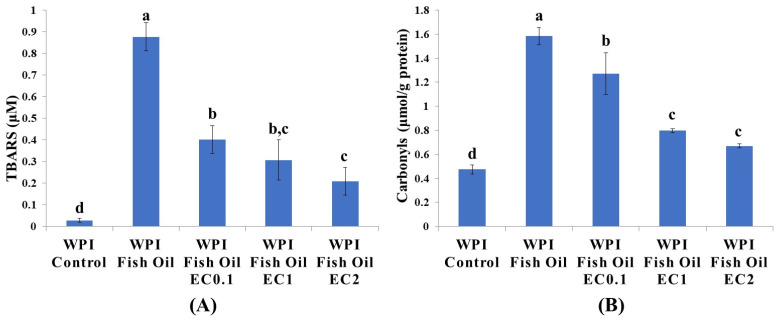
TBARS formation (**A**) and degree of protein carbonylation (**B**) in fish oil enriched WPI emulsions incubated at 25 °C for 24 h in the absence or presence of EC (0.1~2 mM). Means (*n* = 3) of groups with no common letter (a–d) differ significantly (*p* < 0.05).

**Figure 2 foods-12-01559-f002:**
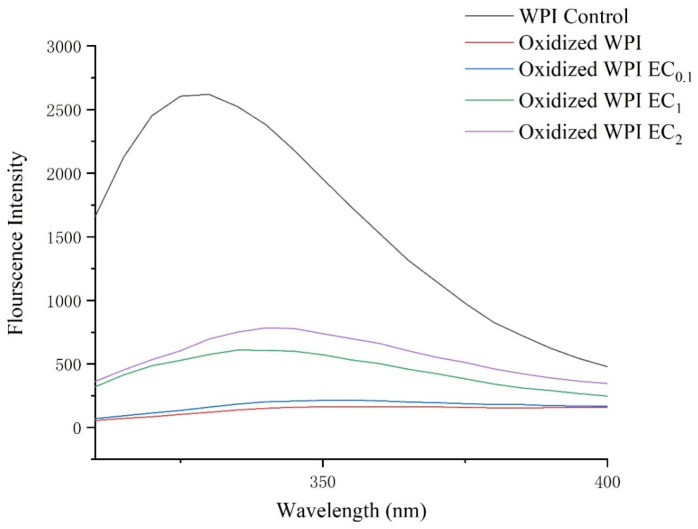
Spectrum of endogenous fluorescence changes in WPI. The fluorescence intensity was record at the excitation wavelength of 295 nm, and emission wavelength ranged from 315 nm to 400 nm.

**Figure 3 foods-12-01559-f003:**
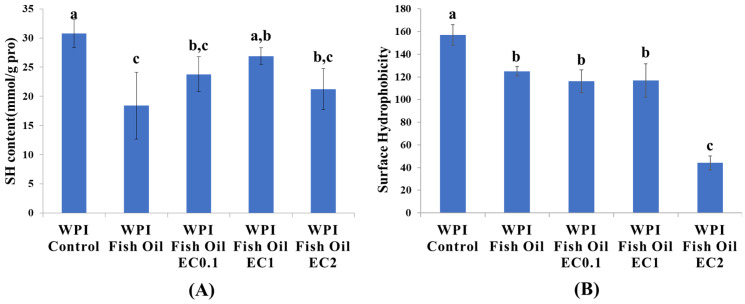
Total SH groups (**A**) and surface hydrophobicity of WPI (**B**) in fish oil-fortified emulsions incubated at 25 °C for 24 h in the absence or presence of EC (0.1~2 mM). Means (*n* = 3) of groups with no common letter (a–c) differ significantly (*p* < 0.05).

**Figure 4 foods-12-01559-f004:**
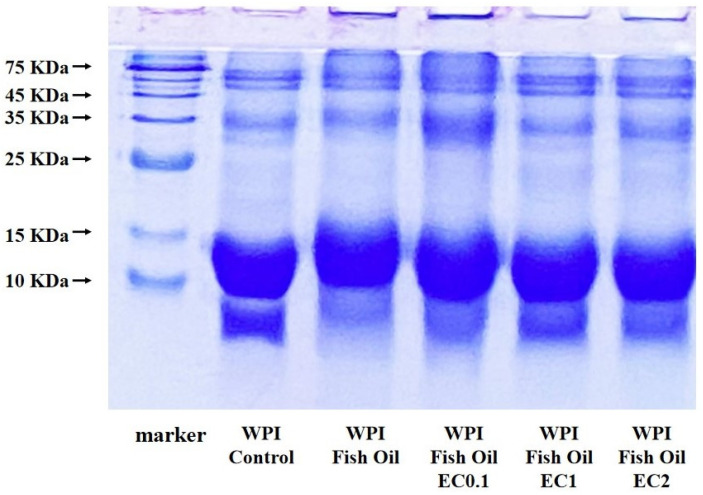
SDS-PAGE analysis of the protein profile of WPI in fish oil-fortified emulsions incubated at 25 °C for 24 h in the absence or presence of EC (0.1~2 mM).

**Figure 5 foods-12-01559-f005:**
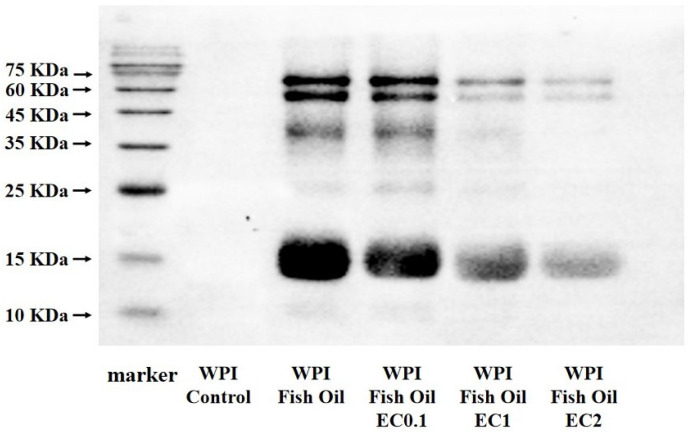
Western bolt analysis of MDA-bound proteins in WPI in fish oil-fortified emulsions incubated at 25 °C for 24 h in the absence or presence of EC (0.1~2 mM).

**Figure 6 foods-12-01559-f006:**
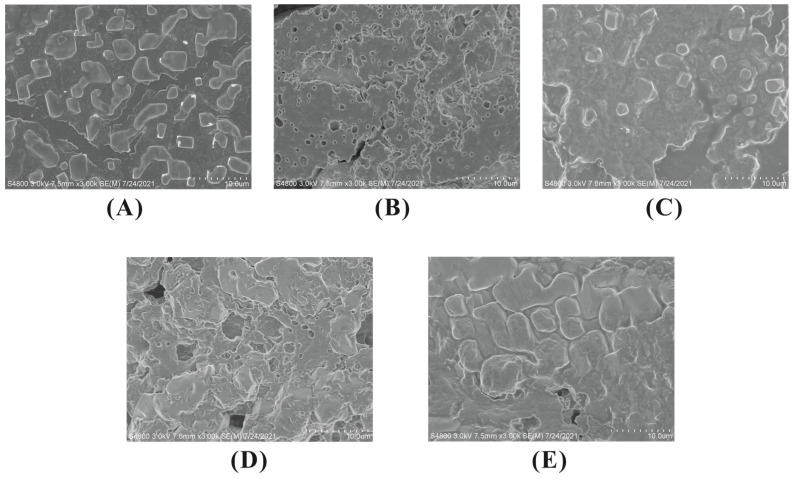
SEM micrographs of WPI (Magnification 1: 3000) in fish oil-fortified emulsions incubated at 25 °C for 24 h in the absence or presence of EC (0.1~2 mM). (**A**) WPI control; (**B**) WPI + fish oil; (**C**) WPI + fish oil + EC 0.1 mM; (**D**) WPI + fish oil + EC 1 mM; (**E**) WPI + fish oil + EC 2 mM.

**Table 1 foods-12-01559-t001:** Protein hydrolysis (%) of WPI in emulsions containing fish oil in the absence and presence of EC (0.1, 1, and 2 mM). Means (*n* = 3) of groups with no common letter in the same columns differ significantly (*p* < 0.05).

Sample	Time (min)			
	Gastric Tract		Intestinal Tract	
	30	60	90	180
WPI Control	4.20 ± 0.3 a	4.51 ± 0.3 a	6.76 ± 1.5 a	13.92 ± 0.4 b
Oxidized WPI	6.52 ± 0.3 a	8.07 ± 0.8 b	9.85 ± 0.3 a,b	8.22 ± 2.7 a
Oxidized WPI EC0.1	6.32 ± 0.5 a	8.01 ± 1.7 b	7.41 ± 0.8 a	11.40 ± 0.6 a
Oxidized WPI EC1	10.82 ± 2.1 b	11.11 ± 2.3 b,c	14.20 ± 2.0 c	12.04 ± 0.7 a
Oxidized WPI EC2	10.03 ± 1.1 b	12.02 ± 1.1 c	13.13 ± 2.4 b,c	14.95 ± 0.7 b

## Data Availability

The data presented in this study are available on request from the corresponding author upon reasonable request.

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
