# Peer review of "Epicatechin Inhibited Lipid Oxidation and Protein Lipoxidation in a Fish Oil-Fortified Dairy Mimicking System"

_foods, 2023, doi:10.3390/foods12071559_

Round 1

Reviewer 1 Report

The main strength of the manuscript is the well-established experimental design and the measurement of many parameters to draw the appropriate conclusions.

The manuscript is clear, relevant for the field and presented in a well-structured manner.
The cited references are mostly recent and within the last 5 years publications (with a few exceptions). It is important to mention that the references before the year 2000 contribute greatly to the explanation of causes and results.
The manuscript is scientifically sound and is the experimental design appropriate.
The figures, the table and the images are appropriate.
They properly show the data and easy to interpret and understand.

The weaknesses of the manuscript include the description of methods chapters.
The results of the manuscript cannot be reproduced in all cases based on the details given in the methods section.
The statistical analysis of the data also needs to be supplemented.
Please supplement the explanation of the results with statistical data and appropriate support.

2.2.
What temperature was used during the production of the oil-in-water emulsion samples? Was the fish oil used in the research completely pure? Did it contain any additives? What kind of container were the samples in during the 24-hour shaking? Why was sodium azide used to suppress microorganisms? This needs to be justified.
The codes of the samples should be entered and clarified in this chapter, as described later, e.g. they are used in the figures. It should be clarified what the WPI Fish Oil sample and WPi Control mean.

2.3.
To what temperature were the samples cooled? How long did that take?
At what speed and for how long were the samples centrifuged?

2.5.
At what speed and for how long was the mixture centrifuged? What units of measurement were the data given in?

2.6.
A more detailed description of the method is required. How much TCA solution is required for a given sample? How long was the precipitation and at the end how much precipitated protein was mixed with how much PBS? What units of measurement were the data given in?

2.8.
The legends of the formula are missing.

3.5.
In line 312, the reference must be corrected according to the regulations (Liu et al., 2011).

It is necessary to standardize the reference to figures in the text, e.g. Figure 1 and Fig.2.

In general, for each method description, it is necessary to specify the unit of measurement that is also used in the presentation of the results.

Reviewer 2 Report

Congratulations on the article. The article notes the advantages of adding antioxidants such as epicatechin to reduce lipid oxidation and subsequent protein lipoxidation in oil-containing dairy products.

Please modify some mistakes:

Lipid peroxidation could have been analyzed by analyzing isoprostanes, which are more specific. 

ine 24, remove bold

Lines 179 and 180, center the formula.

Figure 5. Modify font and size.

Author Response

Response to reviewer 2:

Thank you for your advice.

Detailed notes:

Lipid peroxidation could have been analyzed by analyzing isoprostanes, which are more specific.

Response: Thank you for your advice. Isoprostanes are highly specific indicators of lipid oxidation and thereby are regarded as excellent biomarkers of oxidative stress. They are chemically stable, and easily measured by noninvasive means (ELISA、GC-MS、LC-MS/MS). In this study, we used a classic chromatographic method (TBARS) because it is broad-spectrum for monitoring the general degree of lipid oxidation. We will consider using the new method in future studies.

Line 24, remove bold

Response: It has been changed accordingly in Line 24.

Lines 179 and 180, center the formula.

Response: It has been changed accordingly.

Figure 5. Modify font and size.

Response: It has been changed accordingly.

Reviewer 3 Report

The manuscript entitled Epicatechin inhibited lipid oxidation and protein lipoxidation in a fish oil-fortified dairy mimicking system ID: foods- foods-2294879 is relevant for publication in Foods by MDPI after minor improving.

The article is interesting and inspiring. Well written, understandable and substantively at a good scientific level. It brings new content in the undertaken issues.

Detailed notes:

Line 77: (Douglas Paddon-Jones, 2008)

Is this literature to be cited? If so, it should be the next citation number, but it is not listed in References.

References:

There are two “6” in the position nr 6.

Author Response

Response to reviewer 3:

Thank you for your advice.

Detailed notes:

Line 77: (Douglas Paddon-Jones, 2008)

Is this literature to be cited? If so, it should be the next citation number, but it is not listed in References.

Response: The reference has been checked and corrected in Line 77.

References:

There are two “6” in the position nr 6.

Response: The reference has been checked and corrected in Line 463.